# Aerodynamic Study of a NACA 64418 Rectangular Wing under Forced Pitching Motions

**Dimitris Gkiolas and Dimitrios Mathioulakis ***

School of Mechanical Engineering, National Technical University of Athens, Zografos, 15773 Athens, Greece; gkio.dimitris@gmail.com
* Correspondence: mathew@fluid.mech.ntua.gr

**Abstract:** The aerodynamic behavior of a pitching NACA 64418 rectangular wing was experimentally studied in a subsonic wind tunnel. The wing had a chord c = 0.5 m, a span which covered the distance between the two parallel tunnel walls and an axis of rotation 0.35 c far from the leading edge. Based on pressure distribution and flow visualization, intermittent flow separation (double stall) was revealed near the leading edge suction side when the wing was stationary, at angles higher than $17°$ and Re = $0.5 \times 10^6$. Under pitching oscillations, aerodynamic loads were calculated by integrating the output data of fast responding surface pressure transducers for various mean angles of attack ($\alpha_{m\,(max)} = 15°$), reduced frequencies ($k_{max} = 0.2$) and angle amplitudes $\Delta\alpha$ in the interval $[2°, 8°]$. The impact of the above parameters up to Re = $0.75 \times 10^6$ on the cycle-averaged lift and pitching moment loops is discussed and the cycle aerodynamic damping coefficient is calculated. Moreover, the boundaries of the above parameters are defined for the case that energy is transferred from the flow to the wing (negative aerodynamic damping coefficient), indicating the conditions under which aeroelastic instabilities are probable to occur.

**Keywords:** pitching; double stall; aerodynamic damping; dynamic stall; NACA 64418

## 1. Introduction

Nowadays, there is an ever-increasing need for more efficient power production and cleaner, safer aviation. Modern aerodynamic design is oriented to lighter, high aspect-ratio wings to improve aerodynamic efficiency, which in turn increases structural flexibility [1]. The above trend causes relatively large deformations of the wing structure (resulting in variation of the effective angle of attack) and alters aerodynamic loading.

Time-dependent aerodynamic loading of fixed-wing air vehicles, wind turbine or turbomachinery blades and helicopter rotors occurs on many occasions, either due to the motion of the structure or the unsteadiness of the incoming flow [2]. Under these conditions and at relatively high angles of attack of the airfoil, dynamic stall takes place. The review articles by McCroskey [3,4] are very enlightening, regarding the features of the leading edge vortex shedding and the successive chain of events that govern dynamic stall. Normally, a phase lag appears between the body motion and the aerodynamic forces due to the action of viscosity so that the lift coefficient takes instantaneously much higher values compared to the steady case. Moreover, the lift and pitching moment curves form a hysteresis loop in case of periodic pitching motions due to the alternation of flow separation and reattachment over the surface of the airfoil.

Lee and Gerontakos [5] conducted extensive surface pressure and hot film measurements on a pitching NACA 0012 airfoil under attached conditions as well as during light and deep dynamic stall, analyzing in depth the complex boundary layer events and the stalling mechanisms. Leishman [6] observed, amongst others, dynamic stall to occur via leading edge separation on a NACA 23012 airfoil at low Mach numbers, followed by the shedding of a secondary vortex. The secondary vortex was formed and shed during the

downstroke motion and caused a significant increase in lift. More recently, a detailed survey of various experimental investigations was done in [7] on parameters influencing the aerodynamic forces (reduced frequency, Reynolds number, Mach number, airfoil thickness) of a constant rate pitching airfoil. Geissler and Haselmeyer [8] revealed the important role of transition from laminar to turbulent flow and its evolution during dynamic stall experiments. Their numerical results agreed well with relevant experiments and additionally showed a delay of the onset of dynamic stall in free transition. Three-dimensional effects on the problem have been studied experimentally in [9–11], where a common conclusion is that the tip vortex reduces the relative angle of attack near the wingtip, preventing stall from occurring. Mulleners and Raffel [12] employed proper orthogonal decomposition (POD) in time resolved particle image velocimetry (PIV) measurements and developed a new criterion for characterizing the onset of dynamic stall, correlated with the temporal evolution of one specific POD mode. Zhu and Wang [13] suggest that dynamic stall behavior under pitch oscillations and oscillating freestream, is different in certain conditions and should be modeled separately. Corke and Thomas [14] reported that under certain dynamic stall conditions, unwanted effects may arise such as blade vibration, high stresses, negative aerodynamic damping or aeroelastic instabilities, e.g., stall flutter. Stall flutter occurs at high angles of attack, where the applied aerodynamic forces are nonlinear and not easily predictable [15]. In this case, the amplitude of oscillation is increased in time because of negative torsional aerodynamic damping and eventually a limit cycle oscillation appears [16]. Bhat and Govardhan [17] combined force, moment and PIV measurements in forced pitching oscillations with high mean angles and small amplitude to specify the boundaries of stall flutter for various reduced frequencies. Their conclusion was that the phase lag between the flapping shear layer and the airfoil oscillation, as well as its proximity to the surface, is related to the sign of the aerodynamic damping. From the above discussion, it follows that research on dynamic stall still remains an open issue, due to its poly-parametric nature and certain details, yet need to be addressed.

In the context of the present work, the aerodynamic behavior of a NACA 64418 (a commonly used wind turbine airfoil) wing is experimentally studied under forced pitching oscillations with reduced frequencies up to 0.2, Re up to $0.75 \times 10^6$, mean angles up to $15°$ and angle amplitudes up to $8°$, thus extending from the attached flow regime up to deep dynamic stall conditions. Based on unsteady pressure measurements and flow visualization, the wing aerodynamic loading is studied, revealing details of the flow time evolution, focusing on the lift and moment temporal variation. Moreover, the aerodynamic damping coefficient is calculated, based on which the conditions where flow instabilities may appear are determined, such as stall flutter, when the wing is elastically supported.

## 2. Experimental Setup

The aerodynamic behavior of a pitching rectangular wing, made of aluminum, with a NACA 64418 airfoil section was experimentally examined in a subsonic wind tunnel of the National Technical University of Athens with a free stream turbulence level of 0.2% and flow uniformity of 2%. The chord length of the wing was c = 500 mm and its spanwise length 1390 mm, leaving a gap of 5 mm from the upper and lower wind tunnel walls. The wing was vertically mounted in the 1800 mm × 1400 mm test section (Figure 1a) on two bases located outside the wind tunnel to avoid transferring any vibrations to the tunnel walls due to the wing motion. The center of the spanwise shaft about which the wing performed pitching oscillations was located 0.35 c far from the airfoil leading edge (usually met in horizontal axis wind turbine blades) and it was connected to a stepper motor with a speed reducer, allowing the accurate control of the pitching frequency and angle amplitude (Figure 1b). In the same figure, the plunging mechanism is also shown which was not used in the present work.

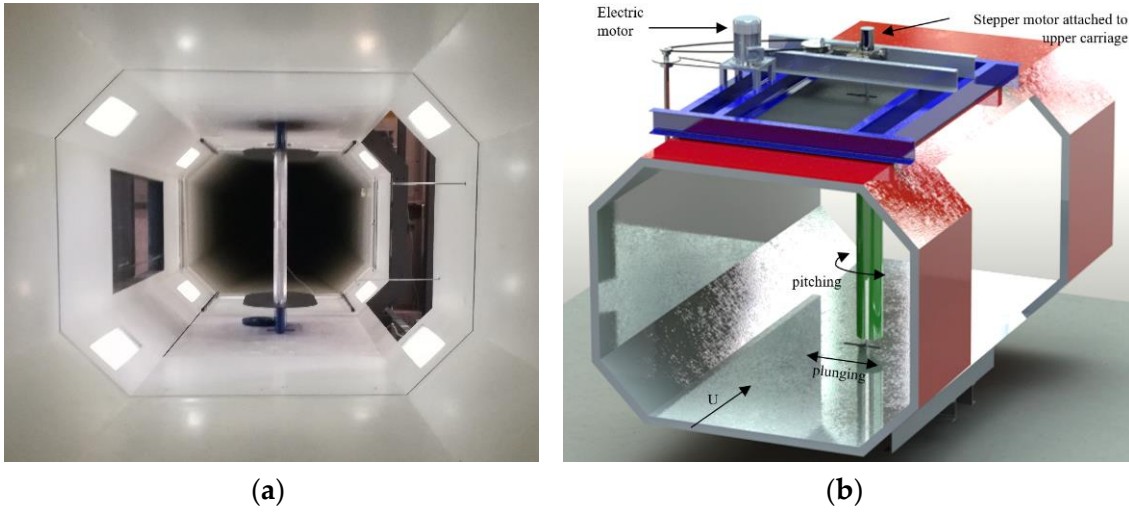

**Figure 1.** (**a**) The wing placed inside the test section; (**b**) a schematic of the drive mechanism of the wing.

The initial set up included the wing with an initial spanwise length of 1000 mm (Figure 2a). However, in order to reduce the tip vortices' effect on the flow field, two aluminum fences 910 mm long were initially attached at the upper and lower end of the wing (Figure 2b,c) which were finally removed by increasing the wing spanwise length up to the wind tunnel walls (see Figure 2d).

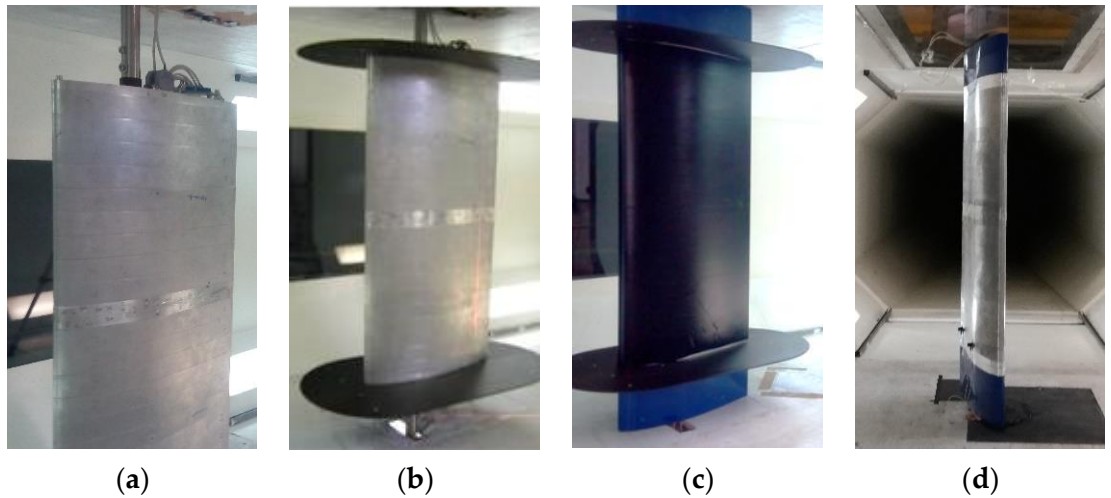

**Figure 2.** Wing model configurations.

The influence of the four wing configurations shown in Figure 2 on the static-lift coefficient angle of attack curve is depicted in Figure 3. More particularly, the 'free ends' case mentioned in Figure 3 corresponds to Figure 2a, the 'fences' case to Figure 2b, the 'fences+ext' to Figure 2c and 'wall2wall' to Figure 2d. The slope of the curve at Re = $0.75 \times 10^6$ for the 'free ends' configuration (Figure 2a) is the smallest (0.0673 deg$^{-1}$), as previously observed in [18], whereas, that with the wall-to-wall case (Figure 2d) is the largest (0.103 deg$^{-1}$), tending to the theoretical value 0.110 deg$^{-1}$ (for an inviscid two-dimensional flow). It is worthy of note, that when the wing span extends to the tunnel walls, the fences practically do not influence the lift. Therefore, it was decided to conduct all the experiments with no fences, using the wall-to-wall wing configuration (Figure 2d).

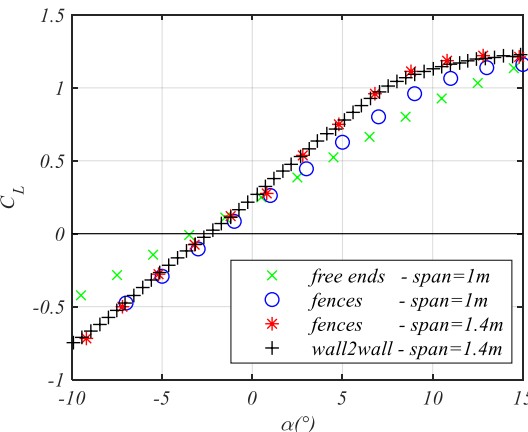

**Figure 3.** Lift coefficient versus angle of attack for various wing configurations. Re = $0.75 \times 10^6$.

The wing was instrumented with 31 piezoresistive pressure transducers (Kulite XCS-062 and Meggitt 8515C-15) (Figure 4a) at the wing midspan, the output of which was amplified by custom made amplifiers, installed in the interior of the wing (Figure 4b). All transducers were calibrated in the interval $\pm 2$ kPa and their sensitivity was close to 1 mV/Pa. The locations of the transducers are shown in Figure 4c, being at small distances between each other in the suction side of the leading edge region where the pressure gradient takes high values.

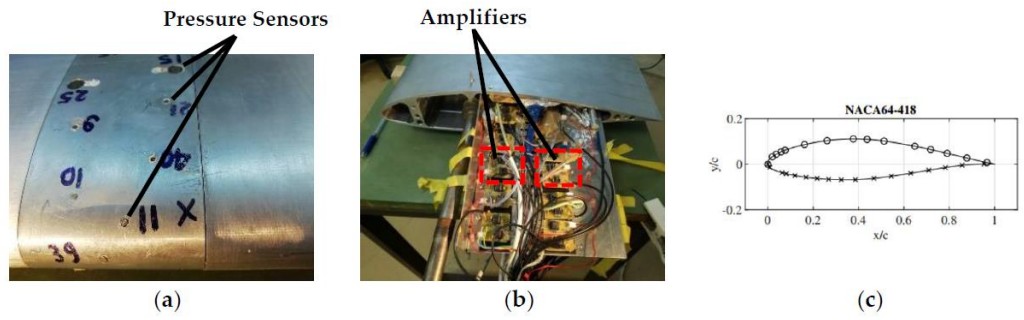

**Figure 4.** (**a**) Pressure transducers at the suction side, (**b**) signal amplifiers in the interior of the wing, (**c**) locations of pressure transducers.

The pressure signals were digitized with a frequency of 400 Hz using a National Instruments A/D card (NI 6031 E) with a multiplexer. The maximum acquisition delay between the first and the last digitized pressure signal is estimated to be of the order of 1ms or less than 0.3% of the minimum period of the oscillations investigated in the present work. The orientation of the wing chord with respect to the free stream was recorded by a wire sensor (with an accuracy of 0.18°), connected with the wing shaft, the output of which was digitized by the above A/D converter. Finally, the total and static pressure of the free stream, two chord lengths upstream of the wing, were recorded versus time using a Pitot–Static tube.

The aerodynamic forces and moments applied on the wing were calculated using the distribution of the pressure coefficient $C_p$ at the midspan region defined as

$$C_p = \frac{p - p_\infty}{\frac{1}{2}\rho U^2} = \frac{p - p_\infty}{p_0 - p_\infty} \tag{1}$$

where $p$ is the static pressure, and $p_\infty$, $\rho$, $U$, $p_0$ are the free stream static pressure, fluid density, velocity and total pressure. Therefore, based on a numerical integration of the $C_p$ values at the midspan region, the lift coefficient $C_l$ and moment coefficient $C_m$ about the quarter-chord point were calculated, assuming that the flow field is two dimensional.

A qualitative picture of the flow field in close proximity to the wing surface was also obtained by both oil flow visualization and a grid of tufts. Oil visualization was applied on the wing suction side covered by a black 90 µm thick flexible film which was painted with a coloured powder mixed with kerosene and lighted with ultraviolet lamps. On the other hand, the tufts were No. 60 sewing thread, 50 mm long, glued on the wing surface in a rectangular grid with a 50 mm step. To avoid any turbulence triggering, the tufts were positioned at distances greater than 100 mm (20% of c) from the leading edge.

## 3. Results and Discussion

### 3.1. Steady Flow Measurements

The pressure distribution at the wing midspan was first recorded under static conditions for various angles of attack in the interval $[-7°, +25°]$ and for two Reynolds numbers, namely $0.5 \times 10^6$ and $0.75 \times 10^6$. Some representative $C_p$—x graphs are shown in Figure 5 where x is the distance from the leading edge along the chord line. It is reminded that after the wing is rotated a certain angle, it stays stationary for a period of two minutes and then pressure data are taken from all sensors for a time interval of 25 s. From each sensor, an average pressure value and its standard deviation were obtained based on which lift and moment coefficients were calculated. The same procedure was repeated at each angle of attack.

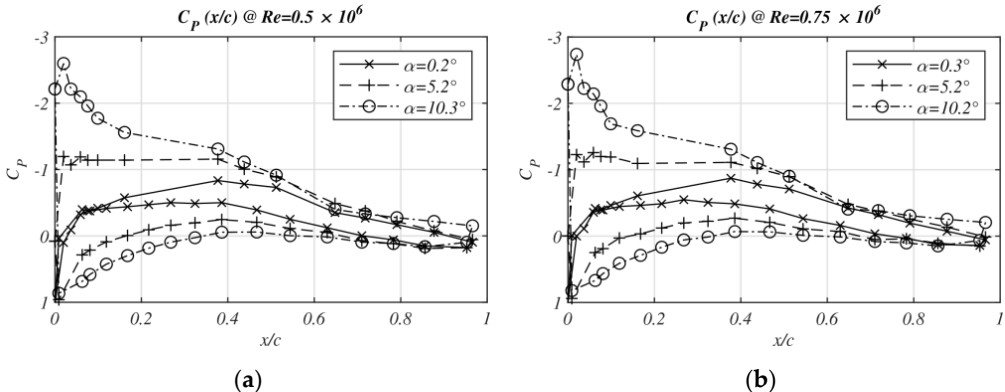

(**a**) (**b**)

**Figure 5.** Pressure distributions at (**a**) Re = $0.5 \times 10^6$ and (**b**) Re = $0.75 \times 10^6$.

It is clear from Figure 5 that for the interval 0 to 10° there were no distinct differences between the pressure distributions of the above two Reynolds numbers, showing the characteristic suction peak at the leading edge region of the wing suction side (at 10°). However, further increasing the angle of attack, differences did appear as is illustrated in Figure 6. Namely, the peak of $C_l$ for Re = $0.5 \times 10^6$ was 1.129 and occurred at 13.8° whereas for Re = $0.75 \times 10^6$, it was 1.184 and appeared at 15.01°. Corrections regarding the effects of solid, wake blockage and streamline curvature are accounted for here, according to references [19,20]. It should be also mentioned that the maximum blockage ratio was 11.74%. Regarding the maximum uncertainty (for a confidence interval of 95%) of the lift and moment coefficients, taking into account both the statistical error and the accuracy of the manometer used for the calibration of the pressure sensors, was estimated to be ±0.015.

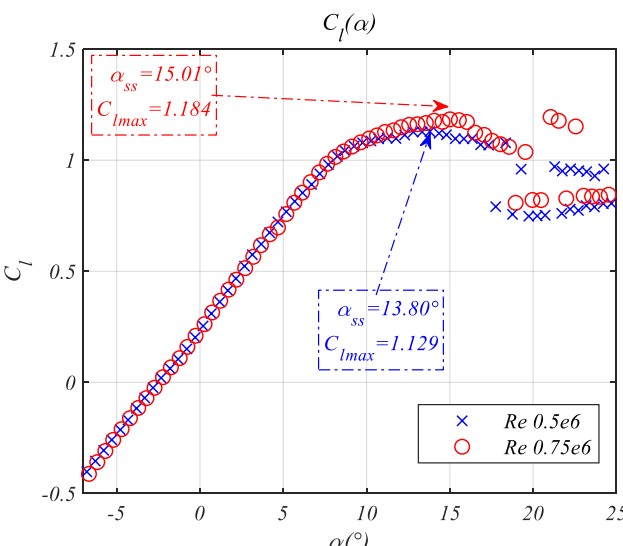

**Figure 6.** Lift coefficient versus angle of attack for Re = $0.5 \times 10^6$ and $0.75 \times 10^6$.

Based on oil flow visualization, it is interesting to notice in Figure 7 that increasing the angle of attack from 11° to 17°, where the lift peak was met, the flow separation line (which deviated from a straight pattern indicating three dimensional flow effects) travelled progressively upstream, covering a distance of 0.2 c. More particularly, at the wing mid span, this line passed from x/c = 0.5 at 11° and reached x/c = 0.3 at 17°. In the latter figure, a straight red line has been drawn at a distance x = 0.4 c from the leading edge. A similar picture was given by the flow visualization based on tufts (see Figure 8) which upstream of the flow separation line were straight and parallel to the free stream whereas downstream of it they were oriented far from the midspan towards the wing ends due to the three dimensional separated flow region. In the same figure (Figure 8), a straight red line is drawn parallel to the leading edge at x = 0.4 c as well as a curved green line which is the separation line identical with that of Figure 7b. It is reminded that Figures 7 and 8 refer to Re = $0.5 \times 10^6$.

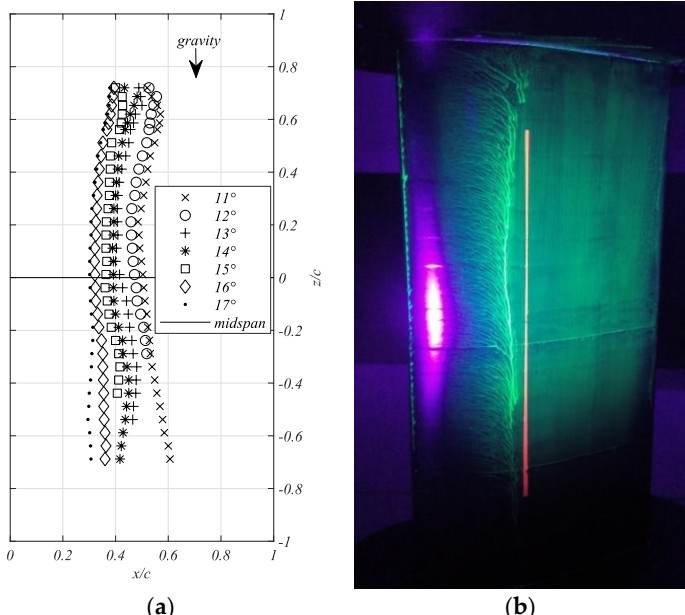

**Figure 7.** Flow separation line based on oil flow visualization. (**a**) Angle of attack from 11° to 17°, (**b**) angle of attack $\alpha$ = 16°. Re = $0.5 \times 10^6$.

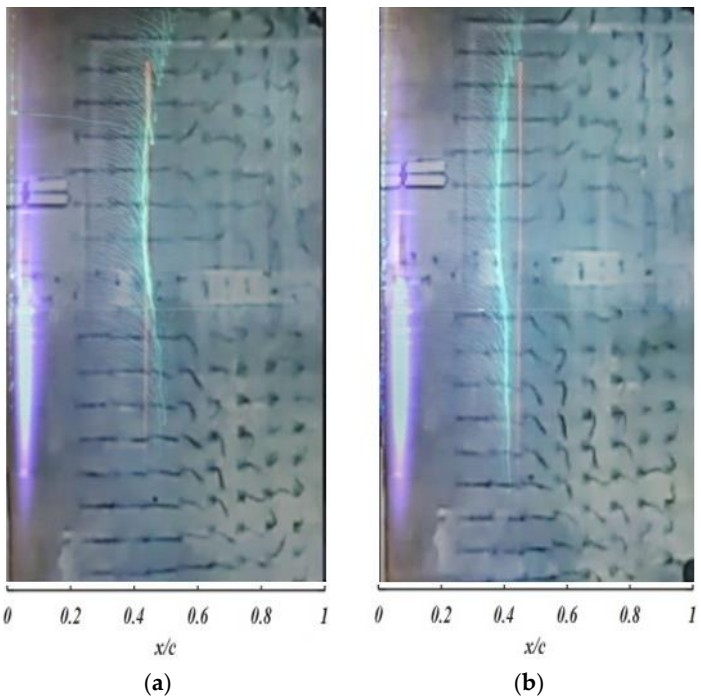

**Figure 8.** Superposition of oil flow and tuft visualization (**a**) $\alpha$ = 14° and (**b**) $\alpha$ = 16°. Re = 0.5 × 10⁶.

More details of the flow evolution are depicted in Figure 9, in which the pressure variation along the chord was shown at 15 points on the wing suction side for angles of attack in the interval $[-5°, +25°]$ and Re = 0.5 × 10⁶. It is reminded that the angle of attack increases with a step of 0.5° following a procedure mentioned in the beginning of Section 3.1. It is characteristic that for $\alpha \geq 17°$, the pressure coefficient in the major part of the chord tends to a constant value of $-0.8$, indicating that the flow has been detached almost from the whole suction side at these angles of attack. Based on the same figure, flow separation was shown to initiate from the trailing edge since the pressure tends to reach a plateau there and progressively moves upstream with an increasing angle of attack.

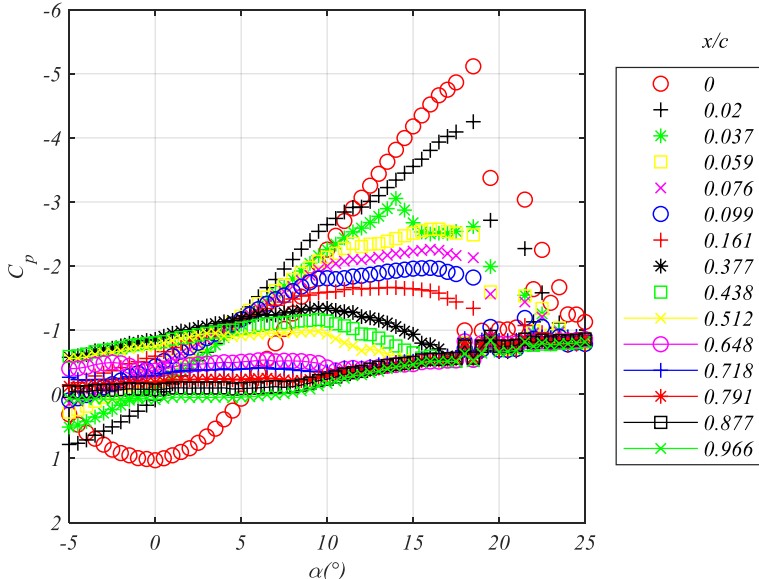

**Figure 9.** Steady case, Re = 0.5 × 10⁶. Pressure at various × locations versus angle of attack.

Another interesting feature of the flow for angles of attack $[17°, 25°]$, is its unsteady character which is depicted by two values of the aerodynamic coefficients for each angle

of attack (see Figure 6). It is reminded that each $C_l$ value shown in Figure 6 was based on 10,000 pressure data for each measurement location, corresponding to a total integration time of 25 s. Increasing the latter time to 120 s, it was realized that at x/c = 0, the histogram of the pressure coefficient consists of two curves, one broader with a peak of about −3 and a narrow one with a peak of about −1 (see Figure 10). The specific time instant of transition between these two values was recorded to occur simultaneously for a series of seven sensors, located near the leading edge on the suction side of the airfoil. This led to the conclusion that the flow alternates for identical inflow conditions from partially attached (lower $C_p$) to fully detached (higher $C_p$). The phenomenon presented here is known in literature as double stall and has been observed to occur in wind tunnel tests, as well as on wind turbine rotors, affecting their performance and power production [21].

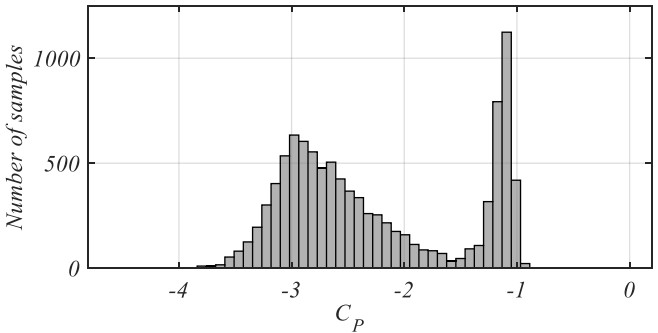

**Figure 10.** Histogram of $C_p$ samples at x/c = 0 for α = 22.5°.

The following two figures (Figures 11 and 12) present a comparison of the results of the current work (NTUA) under static conditions of the lift and moment (about c/4) coefficient with those of three other relevant experimental studies. The data by Abbott and Von Doenhoff [22] include airfoil testing at the NASA Langley two-dimensional low turbulence wind tunnel with wrap around standard roughness over a surface length of 0.08c from the leading edge. In contrast, the presented curves of Ragni and Ferreira [23] and Gonzalez Salcedo [24] (henceforth referred to as CENER experiment) relate to measurements on smooth airfoil surfaces with free transition of the boundary layer. NTUA and CENER measurements were conducted at the same Reynolds number, whereas the other two cases at different ones, greatly affecting the region near stall. The smaller lift slope of the present work compared to Ragni and CENER experiments is attributed to the quality of the surface of the wing which consisted of a number of aluminum blocks with small protuberances. The above surface texture might be a source of flow transition [25].

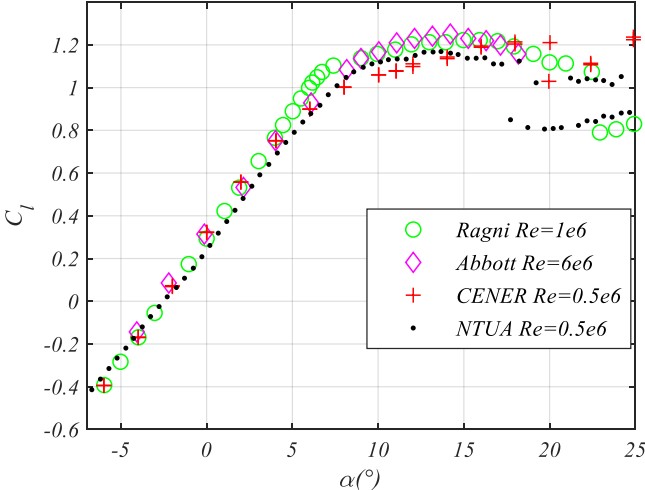

**Figure 11.** $C_l$ versus angle curves. Comparisons with experimental data from literature [22–24].

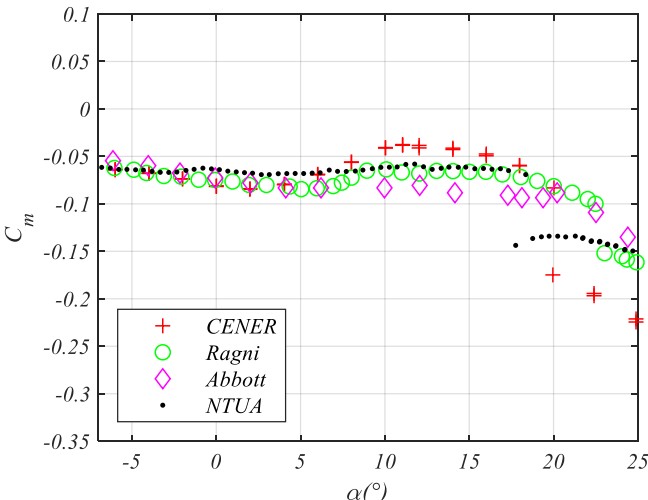

**Figure 12.** $C_m$ curves. Comparisons with experimental data from literature [22–24].

*3.2. Wing Pitching Motion*

The aerodynamic behaviour of the wing undergoing forced pitching oscillations is presented in this paragraph. The mean angle of attack varied in the interval 0° to 15° with an increment of 5°, the geometric angle amplitudes were $\Delta\alpha$ = 2°, 4°, 6° and 8° and the frequency of oscillation varied between 1 Hz and 2 Hz. For the 1 Hz frequency motion, pressure data were recorded for 180 consecutive cycles for each tested case, and 360 cycles for the 2 Hz motion, both at a sampling rate of 400 Hz. Cycle-averaged pressure distributions allowed the calculation of the lift and moment coefficient versus angle of attack loops depicting a representative behaviour of the individual cycles. However, it should be noted that when flow separation is predominant, the cycle-to-cycle variation is not trivial. The data presented below refer to cycle averaged lift and pitching moment coefficients. The continuous line corresponds to the upstroke part of the oscillation (increasing angle of attack) and the dashed line to the corresponding downstroke (decreasing angle of attack). For comparison, the static curve in each figure is shown with a dotted line. The above measurements were performed for Reynolds numbers $0.5 \times 10^6$ and $0.75 \times 10^6$, resulting in a total of 64 cases. However, for brevity, only the results for Re = $0.5 \times 10^6$ will be presented here since the basic trends are similar for the above two Re.

3.2.1. Effect of Mean Angle of Attack

The effect of the mean angle of attack on the flow field is shown in Figure 13 for four different mean angles, $\alpha_m$ = 0°, 5°, 10° and 15° with the amplitude of oscillation varying between 4° (left column) and 8° (right column), the frequency being 1 Hz and the reduced frequency k = $\omega c/2U$ = 0.1 where $\omega$ is the cyclic frequency.

In the attached flow cases (low angles of attack), the curves of the lift and pitching moment present typical elliptical counter clockwise loops. More specifically, at low angles of attack, the lift coefficient loops follow the corresponding static curve although they tend to have lower lift values when $\alpha$ increases (upstroke) and higher when it decreases (downstroke) as Leishman notes in [2]. As the mean angle of attack is increased ($\alpha_m$ = 10° and $\alpha_m$ = 15°), the influence of flow separation becomes stronger. A hysteresis effect causes the lift coefficient loop to change direction of rotation from counter clockwise to clockwise and the moment coefficient to form a clockwise sub-loop. The lift and moment coefficients reach values far beyond their static counterparts (nose down pitching moment), before flow separation region extends, causing lift reduction and a pitching moment increase. Subsequently, flow reattachment occurs nearly throughout the downstroke phase of the wing oscillation until the aerodynamic loads recover. The above process has exactly the characteristics of dynamic stall [4].

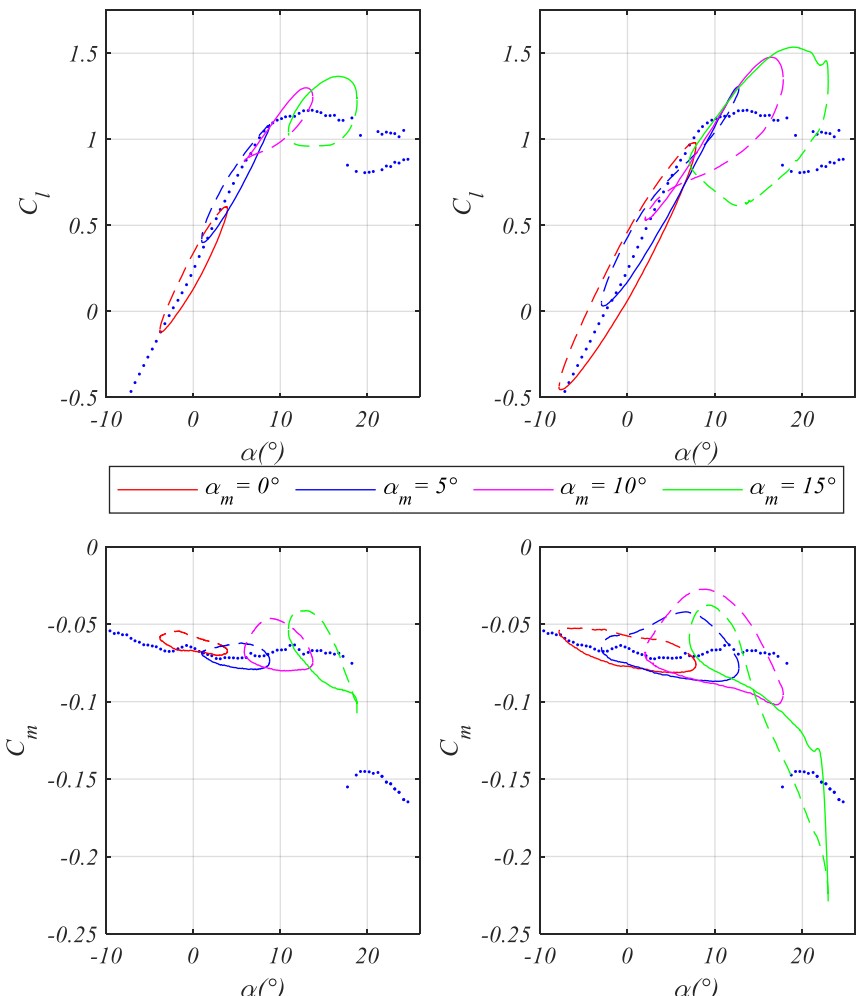

**Figure 13.** Lift and pitching moment, Re = 0.5 × 10⁶, k = 0.1. **Left** column: Δα = 4°, **Right** column: Δα = 8°. Dotted line corresponds to steady case.

In each cycle of oscillation, the maximum variation of the lift and pitching moment coefficients are also calculated. The probability density function of these two quantities, appears to have increased variance when flow separation takes place. These quantities, apart from their importance on the aerodynamic performance of the wing, act also as an indicator of the fatigue of the aerodynamic structure under the above flow conditions. The average values of the maximum peak to peak variation of the lift coefficient $C_{l,p-p}$ in each cycle are shown in Figure 14 for each case as a function of the mean angle of oscillation, angle amplitude and reduced frequency. As the mean angle increases from 0° to 10°, regardless of the angle amplitude or the reduced frequency, $C_{l,p-p}$ gradually decreases. This is explained by the appearance of partial flow separation over the wing, which causes the steady as well as the unsteady measurements to enter the non-linear part of the lift coefficient versus α curve.

If the mean angle is further increased ($\alpha_m = 15°$), $C_{l,p-p}$ no longer depends exclusively on this angle, being influenced by the combination of the angle amplitude and the reduced frequency. Generally, $C_{l,p-p}$ either remains constant or it takes higher values compared to $\alpha_m = 10°$, especially for k = 0.2. This is attributed to the pitch rate during dynamic stall, which allows the boundary layer to remain attached to much higher angles of attack resulting in higher maximum lift coefficients.

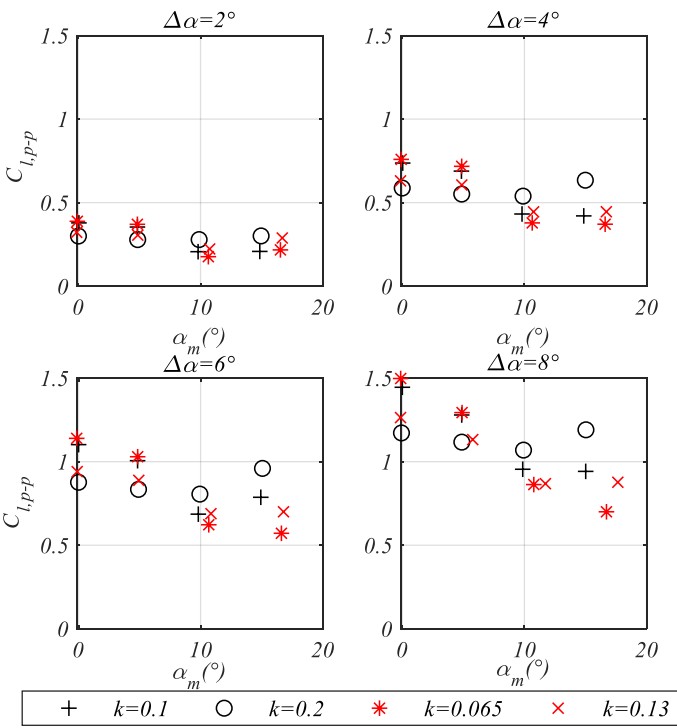

**Figure 14.** Peak-to-peak lift coefficient versus mean angle of attack, angle amplitude and reduced frequency.

In contrast, the peak-to-peak pitching moment coefficient is found to increase as the mean angle of attack becomes higher (Figure 15). It is also worthy of note the particularly large excursions of the moment coefficient at $\alpha_m = 15°$ and for angle amplitudes $6°$ and $8°$ as a result of the dynamic stall process.

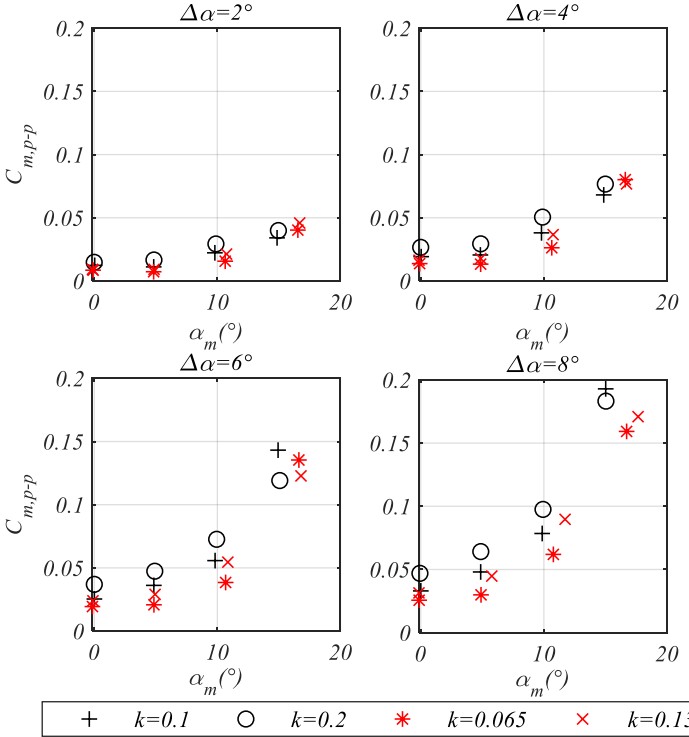

**Figure 15.** Peak-to-peak moment coefficient versus mean angle of attack, angle amplitude and reduced frequency.

3.2.2. Effect of Reduced Frequency

In Figure 16, useful remarks can be made, regarding the effect of the reduced frequency on aerodynamic loads: for angles of attack lower than about 10°, as the reduced frequency k increases, the slope of the $C_l$-$\alpha$ curve is reduced and the loop becomes wider, a phenomenon which is predicted by Theodorsen's theory [26]. As the angle of attack exceeds the linear part of the lift curve, when k is increased, the lift loops tend to change the direction of rotation ($C_l$ taking higher values during upstroke), they become thinner and $C_l$ takes higher peak values.

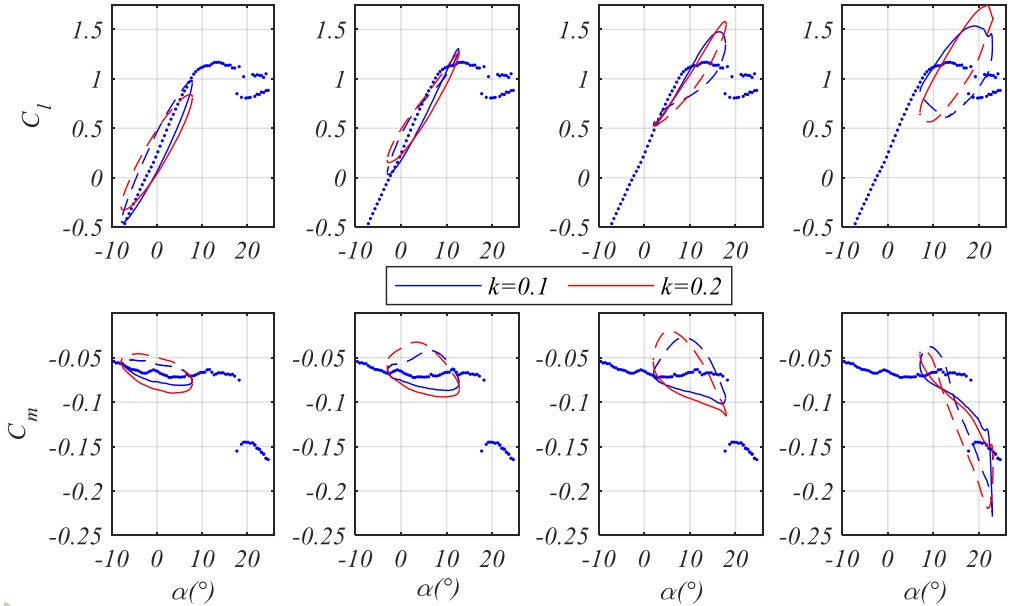

**Figure 16.** Lift and moment coefficients for reduced frequency k = 0.1 (blue loops) and k = 0.2 (red loops).

For the pitching oscillation case of $15° \pm 8°$, the wing operates well above the static stall angle (13.8°) so that dynamic stall takes place. At the higher reduced frequency, the onset of flow separation is delayed compared to the lower frequency, allowing higher $C_l$ values. This is clearly documented by employing the flow visualization with tufts. This technique was applied for two pitching cases ($15° \pm 8°$, k = 0.1 and k = 0.2) by recording the motion of the tufts at a frame rate of 120 Hz, using a led photo-diode as a reference which blinked at a specific angle of attack. In Figure 17a, it can be noticed that for a reduced frequency k = 0.1 and an incidence of 16°, during the upstroke part of the cycle, flow separation covered a significantly larger part of the wing, compared to k = 0.2 (Figure 17b). Comparing the Figure 18a,b with the corresponding static baseline in terms of flow separation extent, we observe the following: for k = 0.1, the image was similar to the static $\alpha = 12°$, while for k = 0.2 it resembled $\alpha = 10°$. Therefore, it can be said that the onset of separation in the case of the lower frequency is delayed about 4° compared to the static measurements and 6° in the higher frequency of oscillation. Of course, this condition favours the flow attachment to the airfoil surface at higher incidence angles affecting the generation of lift, so that a significantly higher $C_{lmax}$ is achieved for the higher oscillation frequency compared to the lower.

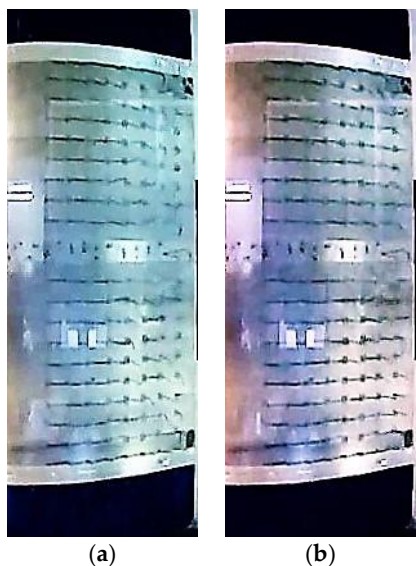

(**a**)                    (**b**)

**Figure 17.** Tuft flow visualization: $15° \pm 8°$: (**a**) 16.0° upstroke, k = 0.1, (**b**) 16.1° upstroke, k = 0.2.

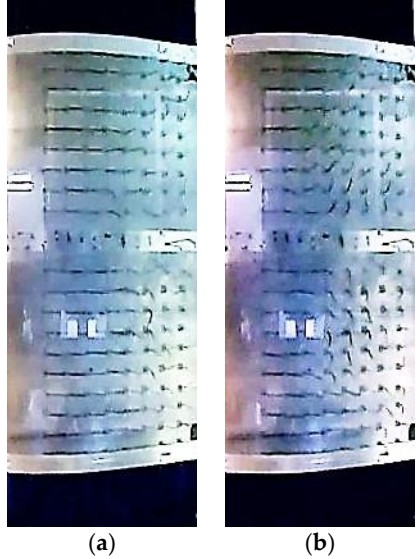

(**a**)                    (**b**)

**Figure 18.** Tuft flow visualization: $15° \pm 8°$: (**a**) 9.9° downstroke, k = 0.1, (**b**) 9.8° downstroke, k = 0.2.

A similar phenomenon occurs during flow re-attachment. For k = 0.1, the extent of the flow separation region at $\alpha = 10°$ is at an intermediate state between the static angles $\alpha = 10°$ and $\alpha = 12°$, while for k = 0.2 it corresponds to the static angle of $\alpha = 14°$. Therefore, it turns out that in the reattachment phase, the flow lags the wing motion similar to the flow separation phase.

### 3.2.3. Impact of Oscillation on Aerodynamic Damping

Whether energy is transferred from the flow to the wing or the opposite, during its pitching motion, the so called energy coefficient $C_e$ is an appropriate index, defined as

$$C_e = C_m \frac{da}{dt} \frac{c}{U} \tag{2}$$

where the moment coefficient $C_m$ and the time derivative of the angle of attack are both positive during upstroke. Since $C_m$ takes always negative values for all the examined cases, the energy coefficient is positive during downstroke which means that during this phase,

energy is transferred from the flow to the wing structure. As shown in Figure 19, increasing the mean angle of attack and the reduced frequency, the positive work progressively increases and eventually exceeds the absolute value of the negative work. A similar conclusion is drawn in [17], which is attributed to the highly non-sinusoidal form of the moment coefficient.

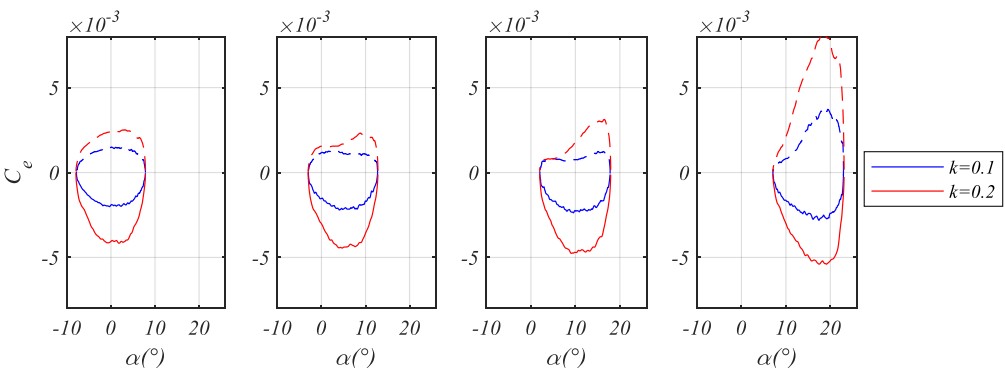

**Figure 19.** Energy coefficient variation in a cycle for various mean angles of attack, two reduced frequencies (0.1 and 0.2) and Re = $0.5 \times 10^5$. Mean angle of attack (from left to right): $\alpha_m = 0°$, $\alpha_m = 5°$, $\alpha_m = 10°$, $\alpha_m = 15°$.

For the case of a single degree-of-freedom harmonically pitching airfoil, the cycle aerodynamic damping coefficient $\Xi$, as defined by Carta and Niebanck [27] and Oates [28], is:

$$\Xi = -\frac{1}{\pi \Delta \alpha^2} \oint C_{m\frac{c}{4}} da = -\frac{1}{\pi \Delta \alpha^2} \int_{\alpha_{min}}^{\alpha_{max}} (C_{m\frac{c}{4}}^{\mathrm{U}} - C_{m\frac{c}{4}}^{\mathrm{D}}) d\alpha \qquad (3)$$

where $\Delta \alpha$ is the angle amplitude and 'U' denotes upstroke while 'D' downstroke. The negative value of the aerodynamic damping coefficient indicates that work is done from the flow to the wing during a cycle which means that in this case, the oscillation is aerodynamically unstable and could lead to aeroelastic instabilities. Indeed, aeroelastic instabilities (self-excited or self-sustained) at high initial angles of attack were observed with the same experimental setup in [29,30], where the airfoil was elastically supported. The probability density function of the above coefficient appears to have increased variance when flow separation is involved. Indicatively, a histogram of $\Xi$ is presented for two cases, one in the attached flow region and the other for a typical dynamic stall case (Figure 20). It is noticeable that the aerodynamic damping coefficient in dynamic stall takes both positive and negative values. For example, the average value of the damping coefficient for a set of cycles may be positive (aerodynamically stable) although in individual cycles this might be negative (unstable).

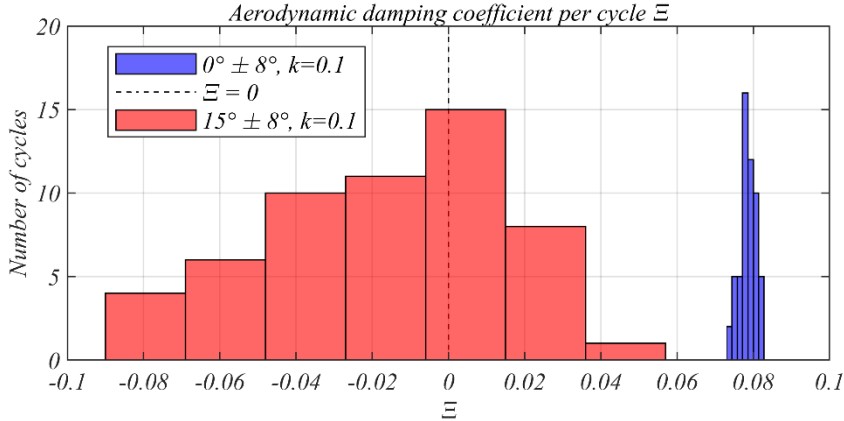

**Figure 20.** Histogram of $\Xi$ of the pitching wing for mean angles 0° and 15°.

Moreover, it can be clearly seen in Figure 21 that for $\alpha_m$ below the static stall angle ($\alpha_m \leq \alpha_{ss}$), the aerodynamic damping coefficient increases almost linearly with an increasing mean angle of attack, similarly to [27]. This is a general conclusion based on the averaged values of the cases studied with constant k and Reynolds number and for different amplitudes of oscillation. When $\alpha_m$ exceeds the static stall value, a more complex mechanism appears, where the aerodynamic damping coefficient acquires negative values (aerodynamically unstable). We can argue that the aerodynamic damping coefficient generally decreases when $\alpha_m \geq \alpha_{ss}$. From Figure 21, however, it cannot be concluded whether the angle $\alpha_m$ is the sufficient condition for transition from a stable to unstable state since the effect of other parameters like k and Re is also apparent. Nevertheless, it is revealed that the existence of negative aerodynamic damping is linked with high mean angles of attack.

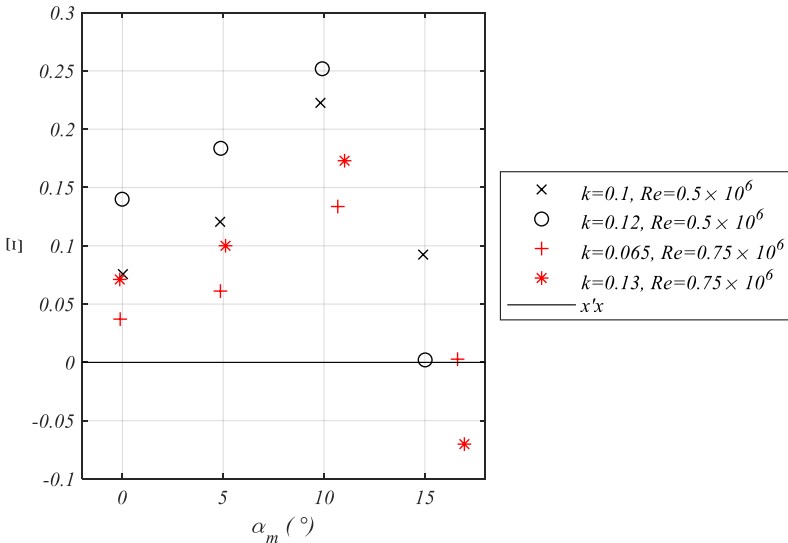

**Figure 21.** Aerodynamic damping coefficient versus mean angle of attack.

Regarding the angle amplitude, there is no indication that the aerodynamic damping coefficient depends on this when the flow is attached (e.g., $\alpha_m = 0°$ or $\alpha_m = 5°$) as shown in Figure 22. However, at $\alpha_m = 10°$ there is a noticeable decrease of this coefficient with an increase of the angle amplitude. In dynamic stall conditions with a mean angle of attack of 15°, the correlation appears to be stronger. This occurs up to a value of $\Delta\alpha = 6°$, where, regardless of Re and k, all the examined cases are aerodynamically unstable. Eventually, at an amplitude of $\Delta\alpha = 8°$, the damping coefficient takes slightly increased values but yet remains negative.

Finally, in Figures 23 and 24, the lift and moment coefficient loops are compared with the corresponding open data set published in the framework of IRPWind European Research Programme (CENER experiment) for one case, namely for a mean pitching angle 15°, angle amplitude 8°, reduced frequency k = 0.1 and Re = $0.5 \times 10^6$. The latter data are very close, both qualitatively and quantitatively with the data of the present work. The slope of the lift coefficient of NTUA during upstroke is observed to be slightly lower than that of CENER. However, the chain of the dynamic stall events, as revealed by the lift loops, occur at nearly the same angles of incidence in the two experiments. Larger differences are found in the pitching moment coefficient, although the similar trends are maintained. The most notable ones are the larger negative peak of the pitching moment and a successive abrupt increase during the flow reattachment process of the CENER experiment.

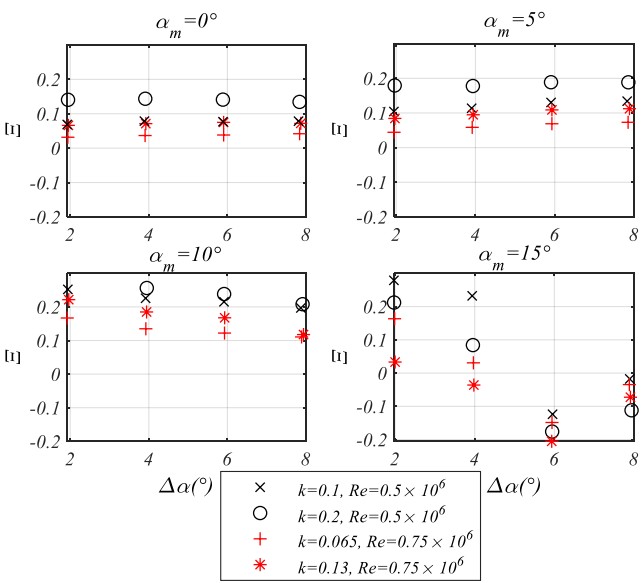

**Figure 22.** Aerodynamic damping coefficient versus amplitude.

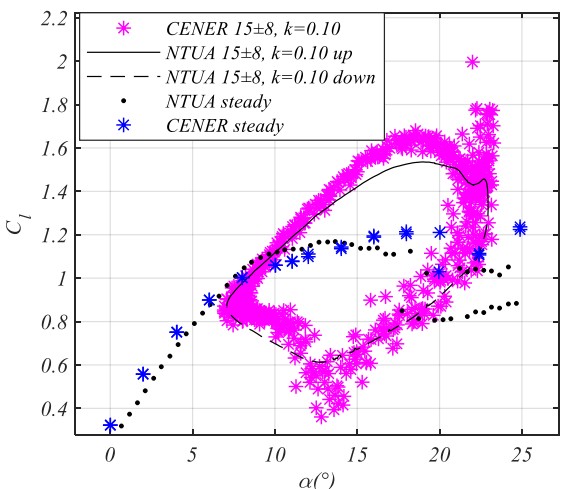

**Figure 23.** Steady and unsteady lift coefficient (mean angle 15°/amplitude 8°). Comparison with literature data [24].

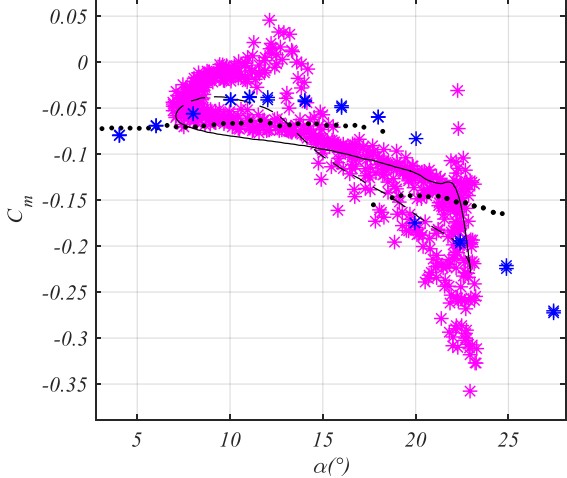

**Figure 24.** Steady and unsteady moment coefficient (mean angle 15°/amplitude 8°). Comparison with literature data [24]. Symbols as in Figure 23.

## 4. Conclusions

The flow about a pitching NACA 64418 rectangular wing was experimentally examined in a subsonic wind tunnel. Having a chord length c = 500 mm and a spanwise length 1390 mm leaving a small gap from the wind tunnel walls, the wing was set to forced pitching motions about an axis 0.35 c far from the leading edge with reduced frequencies up to 0.2, Re up to $0.75 \times 10^6$, mean angles up to 15° and angle amplitudes up to 8°. Based on pressure measurements at the wing midspan region, lift, and moment coefficients were calculated versus the geometric angle of attack.

Under static conditions and for angles higher than 17° (beyond the static stall angle), the lift curve showed two branches due to intermittent flow separation from the leading edge region corresponding to partially attached and separated flow regimes (double stall). On the one hand, during pitching oscillations, the peak-to-peak lift coefficient $C_{l,p-p}$ in a cycle increases with the angle amplitude, whereas for relatively low mean angles ($\alpha_m \leq 10°$), $C_{l,p-p}$ is reduced with increasing mean angle. On the other hand, the peak-to-peak moment coefficient within a cycle is increased with respect to either the mean angle or the angle amplitude. Under dynamic stall conditions, the large excursion of the pitching moment and the shape of its loop are related to negative torsional aerodynamic damping, so that the system becomes aerodynamically unstable. Below the mean angle of 15°, all cases studied in the present work are aerodynamically damped. However, for pitching oscillations about a mean angle near the static stall angle ($\alpha_m = 15°$) and pitching amplitudes from 4° to 8°, negative torsional aerodynamic damping may appear, depending on the reduced frequency and the Reynolds number. Moreover, energy is transferred from the fluid to the wing during the downstroke part of its periodic motion. Nevertheless, under the latter conditions, the sign of the aerodynamic damping coefficient changes between cycles due to the inherently unsteady flow behavior of dynamic stall.

**Author Contributions:** Conceptualization, D.G., D.M.; methodology, D.G., D.M.; software, D.G.; validation, D.G.; formal analysis, D.G.; investigation, D.G.; resources, D.M.; writing—original draft preparation, D.M., D.G.; writing—review and editing, D.M., D.G.; supervision, D.M.; project administration, D.M. All authors have read and agreed to the published version of the manuscript.

**Funding:** This research has been co-financed by the European Union (European Social Fund-ESF) and Greek national funds through the Operational Program "Education and Lifelong Learning" of the National Strategic Reference (NSRF)—Research Funding Program: THALES. Investing in knowledge society through the European Social Fund.

**Conflicts of Interest:** The authors declare no conflict of interest.

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
