# Peer review of "Aerodynamic Study of a NACA 64418 Rectangular Wing under Forced Pitching Motions"

_fluids, doi:10.3390/fluids6110394_

Round 1

Reviewer 1 Report

In this study, the authors performed experimental study in a subsonic wind tunnel to examine the aerodynamic behavior of a selected airfoil during pitching motion.

Please find attached the comments and suggestions for the authors.

Author Response

We would like to thank the reviewer for the constructive criticism and useful suggestions for the improvement of the manuscript.

Below are the answers to the comments (written in bold) of the reviewer.

  1. The paper needs a revision in terms of grammatical errors.

Grammatical errors were checked and corrected.

  1. The paper does not explain clearly the reason why the authors have selected NACA 64418 for their experimental work? Is there any special reason for the selection of this airfoil?

NACA 64418 airfoil is commonly used in wind turbine blades which was added in the revised version. The results of the present work were obtained in the context of a project which was related with the aerodynamic design of large wind turbine blades.  

Also what is the reason to select pitching motions at 0.35 c rather than for example quarter chord point?

The reason is that this is a common practice met in horizontal axis wind turbines.

It is denoted to be calculated about the quarter-chord point (pg. 4. Line 119)

The moment coefficient is calculated about c/4 from the leading edge since this is the standard way in aerodynamic calculations (given the fact that the aerodynamic centre is for all airfoils close to c/4), allowing comparisons with relevant data from existing literature.

  1. What makes the difference between different figures in Figure 2, please explain in the figure itself that each figure description clearly a) Free end configuration etc... ?

Clarifications were provided in the revised version.

Figure 1 and Figure 2 can be merged to explain the experimental setup.

We believe that merging figures 1 and 2, there will be in total five pictures so that it will not be so clear to the reader. 

  1. For fig.3, fences+ext and wall to wall configurations seem to be almost same? How this could be taking into account the existence of fences? The authors should clearly explain the boundary layer formation in the wind tunnel and the effect of fences in the configuration and their effect on the aerodynamic forces.

The following text was added in the revised version:

‘It is worthy to note that when the wing span extends to the tunnel walls, the fences practically do not influence the lift. Therefore, it was decided to conduct all the experiments with no fences, using the wall-to-wall wing configuration (Fig. 2d).’  

Therefore, no fences were finally used. In fact, the wall to wall configuration allowed us to employ the 2D PIV technique in an aeroelastic experiment, which otherwise would not be possible  if the fences were used since they would block the optical access to the wing midspan region (please see our work in [29]).

Are there also Cd versus angle of attack measurements for these cases? How Cl values are calculated? Are these integration of the pressure distribution or do you also have a balance system to measure the aerodynamic forces and moments? It is not clear from the text how the CL values are obtained. Are these values 2d or 3d?

No balance was used. Instead, the aerodynamic forces were based on a numerical integration of the pressure distribution recorded at the mid span region of the wing, assuming that the flow field is two dimensional. The drag coefficient was calculated but it was not included in the manuscript, since it was based only on the pressure distribution (pressure drag). As it is known, the wing drag is measured more accurately through a wake survey.  

The following text was added in the revised version:

‘Therefore, based on a numerical integration of the Cp values at the midspan region, the lift coefficient Cl and moment coefficient Cm about the quarter-chord point were calculated, assuming that the flow field is two dimensional.’

  1. For Figure 4: Please show and explain with arrows and texts the figures shown.

Arrows and text were inserted.

Figure 4a seems to be lower surface of the airfoil.

The image shows the upper surface, namely the suction side of the airfoil.

 What is the reason to have a fine distribution at the leading edge for the transducer positions and coarse one for the lower surface?

There were more densely located pressure transducers near the leading edge since in this region pressure changes with a high rate from point to point (as it is also depicted in the cp –x distributions).  

  1. Figure 5: How the angles of attack are measured? Why the left hand side figure’s angle of attack is 0.2° and right hand side figure’s is 0.2° Please explain also the angle of attack measurement system used in the subsonic wind tunnel and its accuracy/precision or give some references about the system used.

The angle of attack was measured via a linear wire sensor. Regarding the accuracy of the sensor, this is 0.1% of full scale, or 150μm. Since a rotation of the wing by one deg corresponds to a change of the wire length by 840 μm, the accuracy by which the angle is measured is 150/840=0.178°.  Moreover, the analog output of the sensor (55.9 mV/deg) was calibrated by turning the wing by specified angles through a stepper motor with a speed reducer for increased accuracy. Since, the used A/D converter for the digitization of the signal was 16 bit, the resolution of the measured angle was 0.003 deg.

The angles for the two Reynolds numbers of Fig. 5 are indeed slightly different, namely 0.2° versus 0.3° and 10.3° versus 10.2°. Nevertheless, the 0.1° difference of the angle of attack between the above two cases is within the experimental uncertainty.     

  1. Denote the Reynolds number used in Figures 7,8, 9 etc.

Reynolds number was 0.5 x 105. This was added in the text.

  1. Figure 8 is not very clear and the aspect ratio of the figure seems to be distorted as can be also observed from the x axis numbers. Please try to draw a clearer figure probably by applying some image filters also at the background of the figures. Explain Figure 8 in more detail in the text.

The figures were improved. The following text was added:

‘In the same figure (Fig. 8), a straight red line is drawn parallel to the leading edge at x = 0.4 c as well as a curve green line which is the separation line identical with that of Fig. 7b. It is reminded that Fig. 7 and 8 refer to Re = 0.5 x 106.’

  1. It is not also clear how figure 9 is obtained? Is this a pitching motion with a frequency change or the authors performed some steady simulations and averaged in some intervals the Cp values? Please clarify how the measurements are obtained and if it is a time-averaging? What is the time interval used? The measurements should be performed after waiting a specific time when the angles of attack are changed if it is not a pitching motion. (An explanation similar to Figure 6 explanation is needed for Cp measurements)

No, this is the steady case. Namely, the wing rotates a certain angle, it remains stationary for two minutes and during the next 25 secs, the pressure from all sensors is recorded based on which the mean and standard deviation are calculated. The same procedure is repeated till the whole angle of attack range is covered. The above explanations were added in this version. The following text was added:

‘ It is reminded that after the wing is rotated a certain angle, it stays stationary for a period of two minutes and then pressure data are taken from all sensors for a time interval of 25 s. From each sensor, an average pressure value and its standard deviation are obtained based on which pressure, lift and moment coefficients are calculated.’   

  1. Figure 11 and Figure 12 captions should be clearer. Also emphasis cm is calculated at c/4 in the y axis of Figure 12, Figure 13 2nd row etc…

The label of Fig. 11 was changed toCl versus angle curves. Comparisons with experimental data from literature’

The label of Fig. 12 was changed toCm curves. Comparisons with experimental data from literature’.

Also, in the text, the following sentence was added: ‘under static conditions of the lift and moment (about c/4) coefficient’

  1. On pg 9 line 226, emphasize that the amplitude of oscillation is “Da”. It is not clear from

Figure 13 what is Da. (Check also pg 13, line 342)

In the beginning of paragraph 3.2, the symbol of the angle amplitude Δα was added, in this version.  

  1. Please also denote in the label of the Figure 13 that the blue dotted line curves are steady state solutions (are they steady state solutions?)

The following sentence was added in the label of Fig. 13: ‘Dotted line corresponds to the static case’.

  1. Figure 16, Figure 19, Figure 23, Figure 24 etc..: Please write the unit of a [°] on the x axis of the figures.

The symbol of [ °] was added in the above figures.

  1. Figure 18 and Figure 19 are not clear.

We suppose that the reviewer refers to figures 17 and 18 which were made clearer in this version.

  1. Figure 23: In the labels, please also emphasis the reference number of the literature data used to draw the figure

The reference numbers were added in the label of Fig. 23 and 24.

Reviewer 2 Report

  1. More relevant and recent literatures must be included and gaps in the existing literature must be presented in order to establish the objectives and the novelty of the work.
  2. The turbulence intensity, blockage ratio and uncertainty in the measured values must be presented.
  3. In Figs. 4 a and b, either clearer image or schematic diagrams should be presented.
  4. Fig. 6 should also also include cl-alpha curve from other published data for comparison.
  5. Fig. 9 should be Cp vs x/c and alpha as a parameter. I think that is more of the standard practice.
  6. The Cl comparison (Fig. 11) should be done against experimental data published in reputed journal, at same Re and turbulent intensity. 
  7. Lift Comparison should also be done for pitching case as well.
  8. The quadrature used to compute aerodynamic loads must be presented.
  9. The analysis of the flow physics behind the findings presented must be included in the text.
  10. Conclusions can not be drawn on the effect of reduced frequency just on the basis of two values of the parameter.
  11. Conclusion should be more compact and must include important  quantitative findings as well.
  12. Conclusion should only include novel findings from the current work.

Author Response

We would like to thank the reviewer for the constructive criticism and useful suggestions for the improvement of the manuscript.

Below are the answers to the comments (written in bold) of the reviewer.

  1. More relevant and recent literatures must be included and gaps in the existing literature must be presented in order to establish the objectives and the novelty of the work.

The ‘Introduction’ paragraph was rewritten following the suggestion of the reviewer.

  1. The turbulence intensity, blockage ratio and uncertainty in the measured values must be presented.

Free stream turbulence intensity was 0.2%, maximum blockage ratio 11.74% and maximum uncertainty (95% confidence level) of both the lift and moment coefficients is estimated to be ±0.015, based on the statistical uncertainty of the recorded pressure data as well as the accuracy of the pressure transducer used in the calibration of the pressure transducers.  

  1. In Figs. 4 a and b, either clearer image or schematic diagrams should be presented.

More clarifications are provided by inserting arrows and text in the images.

  1. Fig. 6 should also include cl-alpha curve from other published data for comparison.

Comparisons are made in Fig. 11 regarding the lift coefficient and in 12 for the moment coefficient. It was decided to present in Fig. 6 only the data of the present work in order to be easier for the reader to detect the relevant details mentioned in the text.

  1. Fig. 9 should be Cp vs x/c and alpha as a parameter. I think that is more of the standard practice.

We agree that Cp-x is the standard practice. However, it was decided to show the results in a more compact way, as it is given in Fig. 9. Otherwise, there would be too many figures, so that it would not be convenient for the reader to realize the pressure distribution as a function of the angle of attack.

  1. The Cl comparison (Fig. 11) should be done against experimental data published in reputed journal, at same Re and turbulent intensity.

The data for comparison were extracted from reputable publications. For example, Abbott’s work at NASA is considered one of the most reputable aerodynamic data base. Of course, it should be mentioned that it Is not an easy task to find data under exactly the same experimental conditions as those of the present work.  

  1. Lift Comparison should also be done for pitching case as well.

This is done in Fig.23. It has to be stressed that it is not easy to find a publication with exactly the same experimental conditions.    

  1. The quadrature used to compute aerodynamic loads must be presented.

The lift was calculated by integrating the pressure data at the wing midspan region, assuming that the flow field is two dimensional.

  1. The analysis of the flow physics behind the findings presented must be included in the text.

Based on the pressure distribution, the physics of the flow is presented mentioning the important influence of viscosity in the unsteady flow field, characterized by a phase lag between the body motion and flow separation.

  1. Conclusions cannot be drawn on the effect of reduced frequency just on the basis of two values of the parameter.

In fact, there were examined four reduced frequencies ranging from 0.065 to 0.2 (please see Fig. 14 and 15).

  1. Conclusion should be more compact and must include important quantitative findings as well.
  2. Conclusion should only include novel findings from the current work.

The ‘Conclusions’ paragraph was rewritten mentioning the most important and novel findings of this work.

Round 2

Reviewer 1 Report

Thank you to the authors for the revisions provided. The paper can be published with this revised version by taking into account the below comment.

Just a comment about figure 4a. I am not still sure that the figure represent the suction side of the airfoil since the pressure transducers at the leading edge are not visible from the figure. I would suggest to authors to check this figure or to denote the reason why there are not pressure transducers located at the leading edge of the airfoil although it is denoted that they exist and they are even denser to measure the suction pressure peak.

Author Response

We would like to thank the reviewer for his comment.  

 comment about figure 4a. I am not still sure that the figure represent the suction side of the airfoil since the pressure transducers at the leading edge are not visible from the figure. I would suggest to authors to check this figure or to denote the reason why there are not pressure transducers located at the leading edge of the airfoil although it is denoted that they exist and they are even denser to measure the suction pressure peak.

Answer

Yes, it is the suction side of the airfoil. Since at the airfoil leading edge the surface is curved, the much smaller Kulite sensors (diameter 1.7 mm) were installed, instead of the much wider (diameter 6.35 mm) MEGITT sensors which were used in the rest part of the airfoil. Due the small size of the Kulite sensors, they were not visible in Fig. 4a. Therefore, a new picture was taken and attached in this second revision of the manuscript, showing both types of sensors (Kulite and Meggitt).    

Reviewer 2 Report

No comments

Author Response

We noticed that after submitting our answers, the reviewer does not provide any comments. However, although in the initial review, the item 'are the methods adequetly described?' was characterized by the reviewer as 'can be improved', after the revision he suggests that this must be improved, namely for some reason the new version in this respect is worse than the original, without providing any specific comments. The same holds for the item 'are the results clearly presented?'.  Therefore, we believe it would be important if the reviewer could be more specific in explaining what specific points have to be improved in the manuscript. 

Thank you.